# Co-Producing Paws on Campus: A Psychoeducational Dog-Facilitated Programme for University Students Experiencing Mental Health Difficulties

**DOI:** 10.3390/ijerph21081066

**Published:** 2024-08-14

**Authors:** Joanne M. Williams, Jillian Bradfield, Andrew Gardiner, Patricia Pendry, Laura Wauthier

**Affiliations:** 1Department of Clinical and Health Psychology, University of Edinburgh, Edinburgh EH8 9AG, UKlaura.wauthier@worc.ox.ac.uk (L.W.); 2The Royal (Dick) School of Veterinary Studies, University of Edinburgh, Midlothian EH25 9RG, UK; andrew.gardiner@ed.ac.uk; 3Department of Human Development, Washington State University, Pullman, WA 99164, USA; ppendry@wsu.edu

**Keywords:** animal-assisted interventions, students, mental health, dogs, human-animal interaction

## Abstract

Declining student mental health is a global public health issue. Campus-based animal-assisted interventions (AAIs) are popular and effective interventions to prevent and alleviate symptoms. How to design, implement and evaluate evidence-based, student-centred interventions that enjoy sustained stakeholder buy-in and support is less known. This paper presents the procedures and results of a three-stage co-production method and the resulting curriculum of a novel AAI aimed at university students experiencing serious mental health problems. Stage 1 shaped the focus and structure of the intervention based on online student surveying (N = 204) and consultations with stakeholders (N = 10), including representatives of Student Well-being Services leadership, veterinarians, animal welfare charities and Therapets volunteers. In Stage 2, we conducted co-production workshops with post-graduate students (N = 6), developing the curriculum based on Stage 1 insights. In Stage 3, through iterative prototyping and student feedback (N = 22) the Paws on Campus programme was finalised, resulting in a series of four, one-hour themed sessions: (1) Thoughts and Feelings, (2) Well-being and Welfare, (3) Care and Compassion and (4) Problem Solving and Help Seeking. We describe the co-production method and resulting programme characteristics and provide considerations for others interested in developing effective and sustainable AAIs for their respective populations and contexts.

## 1. Introduction

### 1.1. Increasing Prevalence of University Student Mental Health Problems

The prevalence of mental health difficulties among university students is rising, posing a global public mental health challenge. According to the World Health Organization (WHO) World Mental Health Surveys, the 12-month prevalence of any mental illness increased from 20.3% among university students in 2016 [1] to 31.4% in 2018 [2]. Most recently, Lipson et al. (2022) [3] reported that by 2021, 60% of US students met the criteria for one or more mental health problems, an increase of 50% since 2013. In the UK, too, the number of undergraduate students reporting a mental health condition has increased 2.5-fold between 2013 and 2018 [4]. This is likely to be the tip of the iceberg, as 53% of UK undergraduate and postgraduate students who claimed a mental health condition, according to a 2019 Unite survey, did not disclose this to their university [5]. Beyond the concerns for student well-being from a humanistic perspective, the increasing prevalence of mental health issues is of great concern to university administrators due to their associations with poor grades, low class attendance, difficulties coping with the academic load and social isolation, as well as an increased likelihood of dropping out without graduating [6,7]. 

### 1.2. Systematic and Integrated Programme Development 

In view of these concerns, there have been calls for a systematic and integrated approach to the development of prevention programmes and efforts to promote student well-being. For example, Duffy et al. (2019) [8] promote the use of a triage framework at point-of-first-contact staffed by experienced mental health clinicians who can facilitate transitions to different levels of care when clinically indicated. Unfortunately, many universities do not have the capacity to provide such resources for the increasing number of students who need them, and, as such, are eager to implement lower-cost solutions [9] provided by staff or volunteers without clinical qualifications to provide well-being services. This includes university-based Animal-Assisted Interventions (AAIs), which have seen tremendous growth in popularity and prevalence both in the US [10] and in the UK [11]. Moreover, due to their popularity with students and administrators alike, the evidence base on their implementation and efficacy within university contexts has grown significantly. AAI practice has grown faster than the research evidence and, as complex interventions, AAIs can present implementation challenges that influence their efficacy. Managing the risks to the students and animals involved is vital, and developing programmes that can be sustained over time is important. These developments in AAIs have led to an increase in available resources for university administrators as they explore solutions to an urgent problem. 

### 1.3. Facilitating Sustained Implementation: Student and Stakeholder Buy-In 

A review of the AAI evidence base reveals that administrators must consider several factors to ensure effective implementation. For example, AAI approaches range widely from one-off visits of animals for students to interact with at times of heightened stress, such as exam times [12,13], to a series of regular sessions to support student learning and foster academic stress management skills [14,15], to permanent drop-in centres where students can interact with dogs on an ongoing informal basis [16]. Despite the heterogeneity in programmes and target populations, reviews of efficacy trials [17,18] show a range of positive and mixed results, suggesting overall positive effects on stress- and mental health-related symptoms. However, given the wide range of featured approaches, populations and outcomes examined, university administrators and interventionists still find it challenging to select programme and evaluation approaches suitable for their student population and implementation context. In addition to the need to develop AAIs that are responsive to the individual needs and preferences of students, administrators must consider the contextual factors associated with programme implementation success, including the availability of financial, professional and instrumental resources. To facilitate sustained implementation and integration with well-being and mental health services, it is essential to secure buy-in from all stakeholders (e.g., university students, student well-being support staff, academic staff, and animal welfare experts), who are, in actuality, rarely considered in the design of campus-based AAIs. 

### 1.4. Co-Producing a University AAI with Students and Stakeholders

Several authors have suggested that engaging with patients, clients and stakeholders during the development of mental health interventions is essential to ensure that interventions meet their needs, are sustainable and become integrated in the existing implementation context. For example, within health research, it has become common practice to use a Patient and Public Involvement (PPI) approach with adults, and this has recently also been deemed essential and ethical to implement with children and young people [19]. PPI engages patients in planning, designing and evaluating healthcare interventions to ensure that healthcare interventions are patient-centred, responsive to individual needs and foster engagement and empowerment among patients. For university AAI interventions to be efficient, effective and acceptable to students, it is important to include students in co-producing such interventions to ensure they meet their needs. Similarly, considering the needs and resources of other university stakeholders is essential to secure their initial buy-in, commitment to sustained implementation and eventual integration into their implementation context. Co-production of interventions with stakeholders can achieve this by involving the service users working in partnership with the service providers to develop a service or intervention [20]. 

### 1.5. Rationale for Co-Production 

There are several papers highlighting the emerging use of co-production in public health intervention designs [21] as well as its use as an emerging method of developing school-based health interventions [22]. For example, authors have highlighted the role of various types of stakeholders that have been engaged in the process of co-production including staff members, teachers, principals, senior management, parents and external community representatives. Within the field of AAI, however, co-production approaches are still rare; in fact, a review by these authors found only one paper that featured a co-production approach within the AAI literature, which engaged teachers to successfully create a ‘reading to dogs’ intervention for children [23] and subsequently led to a successful feasibility study [24]. To our knowledge, there are currently no published papers describing the process of co-producing AAI interventions targeting university students’ mental health. 

### 1.6. The Current Study

This paper thus describes the process and outcomes of co-producing a university canine-facilitated mental health intervention for students experiencing mental health difficulties and those who are especially vulnerable within university contexts. Taking a One Health approach, we aimed to ensure the intervention meets the needs of vulnerable students, safeguards the students and the dogs involved and fits within other student support systems by including various university stakeholders. Consequently, a co-production process was adopted involving students, university student well-being staff and counselling services, veterinary and clinical psychology faculty, representatives of animal welfare organizations and volunteer animal handlers. 

Guided by established models for co-producing public health interventions [22], this paper describes a mixed-methods, three-stage process of co-production of a novel university AAI programme, ‘Paws on Campus’. Our co-production activities and approaches were embedded within the 6SQuID method for developing quality interventions [25], which has been used widely to create public health interventions to support young people’s behaviour [26] and health [27]. Within this broad approach, we engaged in a range of co-production activities with students and other key stakeholders that have been used in developing mental health interventions for adolescents and emerging adults [19,28]. As the first co-produced university AAI, this paper outlines procedures and outcomes for co-producing AAI mental health interventions, to ensure they are evidenced-based, enjoy responsiveness from a variety of stakeholders and effectively meet the needs of vulnerable university students and canine partners.

## 2. Materials and Methods

### 2.1. Design 

A three-stage mixed methods approach was adopted to co-produce the content, intervention materials and delivery methods of Paws on Campus. The three stages were as follows: (1) stakeholder consultations (a. student survey, b. interviews with student well-being staff and c. interviews with animal welfare professionals); (2) co-production workshops with post-graduate students and (3) prototyping Paws on Campus and student feedback.

### 2.2. Ethics 

The survey and co-production/feedback work was approved by the University of Edinburgh Clinical and Health Psychology Research Ethics Committee (CLIN720 and 22-23CLPS111, respectively). 

### 2.3. Stage 1: Stakeholder Consultations

#### 2.3.1. Survey of Students AAI Experience and Preferences

A purposive sample of 204 participants completed an online survey via Qualtrics, a secure online survey platform (see Table 1 for demographics). Participants were recruited via email and social media over a period of five months (May–September 2020) through various university student group forums and human–animal interaction research organisations. Inclusion criteria comprised being at least 18 years of age and being a current university student (undergraduate or post-graduate). 

The survey instrument included questions on demographics, a range of measures regarding pet ownership and relationships and a series of nine questions relating to AAI familiarity and preferences, which are reported here. These items were designed to reveal what university students thought of AAI events, whether they had previously attended one at their university and what they are looking for regarding future AAI events. Questions included what animals they wanted to interact with, how often they would attend a session, what activities they would be interested in engaging in during an AAI session, what benefits they expected from attending an AAI session and for what reasons an AAI session would be helpful for them. 

#### 2.3.2. Interviews and Consultations with Student Well-Being Staff

A purposive self-selected sample of representatives of Student Well-being Services included Well-being Advisor leadership, student counselling representatives and disability support leads. The university the study was undertaken at already runs an annual dog visitation programme for students during exams called ‘Paws against Stress’ and key staff involved in the non-targeted AAI supported the development of this more intensive targeted programme, designed specifically for students experiencing mental health difficulties. A series of in-person and online meetings took place with leadership of the university Student Support and Well-being Services over a period of months. This was followed up with a Teams meeting with Well-being Advisors from across the university to discuss the referral process and referral documentation. While interviews were not recorded, given their exploratory nature and the aim to establish collaborations for the project, extensive notes were taken afterwards and stored securely along with email correspondence.

#### 2.3.3. Interviews and Consultations with Animal Welfare Professionals

To ensure that the programme was also cognisant of animal welfare, we designed the programme in consultation with a purposive sample of animal welfare experts including veterinarians (e.g., Dr Andrew Gardiner), Canine Concern Scotland, Edinburgh, Scotland who offer a canine Therapets service (CEO, area representative and volunteer handlers), and the Scottish SPCA (Director of Innovation and Strategic Partnerships) to oversee canine welfare. As a One Health intervention, it was vital to ensure the welfare of the dogs and to view them as partners in the intervention rather than as a ‘resource’. Key outcomes from this process are presented in Results. 

### 2.4. Stage 2: Student Co-Production of Intervention 

A purposive self-selected sample of six postgraduate students who were enrolled in the MSc Psychology of Mental Health (Conversion) programme engaged in four two-hour co-production workshops to develop and refine the intervention structure and materials. Using Stage 1 findings and research on university-based AAIs, student mental health and psychological interventions, students were presented with example materials to focus their discussions and invited to arrive at key points of consensus on the design of the programme. Consensus points were recorded to inform programme design. 

#### 2.4.1. Workshop 1: Theory and Goals

In the first workshop, the six post-graduate students were introduced to co-production, the plan for sessions and some basic theory on AAI. They were then invited to discuss two main topics: (1) identifying some student-focused problems; and (2) their views on how AAIs might help. Afterwards, students discussed these topics in small groups, brainstorming what they thought a university AAI should look like using a worksheet as a prompt guide. Data were collected by drafting ideas on a flipchart (which was then photographed), and worksheets were collected at the end. 

#### 2.4.2. Workshop 2: Activities and Themes

In the second workshop, the facilitator and two Therapets volunteers brought their therapy dogs (three dogs) so that students could trial activities and provide feedback on them. The six post-graduate students were provided with a list of six possible activities, and each small group had time to trial two or three of the activities. Students completed a worksheet rating how each activity went, how much they enjoyed the activity and what they thought could be improved or changed. Students were then given an opportunity to brainstorm a new activity themselves, try it and reflect on how it went. This derived a set of human–animal interaction activities for the programme.

#### 2.4.3. Workshop 3: Content, Order and Layout

This workshop focused on the logistics of the programme, number and order of sessions, session themes and content. The six post-graduate students brainstormed this in small groups on worksheets, and it was then discussed as a larger group. The session also explored how the available space might be set up in a way that felt comfortable for students, dogs and volunteers and identified the materials needed. This was photographed and documented once a suitable layout was found. 

#### 2.4.4. Workshop 4: Trialling and Outdoor Session

The final workshop had been left open so that the six post-graduate students could decide what would be most helpful to discuss. Based on the results of the previous workshops, it was decided that it would be useful to trial a full session, and to explore the feasibility of an outdoor session. This workshop provided feedback on some of the intervention materials being designed, especially the session booklets. 

### 2.5. Stage 3: Prototyping and Student Feedback

In Stage 3, a draft intervention manual and facilitators guide were produced, along with a referral form for the Student Well-being Advisors, session materials and booklets the for students, and pilot evaluation and feedback materials (e.g., pre- and post-session momentary assessments [happiness, anxiety, depression and stress], post-session feedback questions and post-intervention interviews) to gather feedback. All aspects of the programme materials underwent rigorous ethical review and approval by the Department of Clinical and Health Psychology and the Royal (Dick) School of Veterinary Studies. 

A purposive self-selected sample of 17 undergraduate student volunteers from veterinary studies and psychology engaged in pilot sessions and provided written feedback. A further small sample of seven students referred to the programme by Well-being Advisors engaged in a trial of the full programme and completed mood checks before and after sessions, and five of these participants engaged in individual interviews about the programme. Mood check and interview data were analysed and are presented below.

## 3. Results

The three-stage co-production process to develop Paws on Campus took 36 months to complete, from 2020 to 2023 (with a pause in work due to COVID-19 lockdowns and moves away from campus). Co-production involved nine main activities, and the process was iterative and cumulative. Figure 1 illustrates the key processes involved. This section will present the findings for each element of the co-production process, and a summary table is included at the end of the section to synthesise the key design features of the programme. 

### 3.1. Stage 1: Stakeholder Consultations

#### 3.1.1. Student Survey Findings 

There was a high degree of positivity and interest among students in relation to university campus-based AAIs, with only 9.8% saying they would not attend AAIs, even though 64% of participants had no previous experience with AAIs. The preferred animal for AAI was dogs, with 91% of student indicating this preference. In terms of the setting, the preferred option was small groups of students with animals (64%) and a series of sessions held either monthly or weekly. Seventy-seven percent of students stated that they would prefer sessions in the afternoons. Table 2 shows the descriptive results from the student survey on their preferences for campus-based AAI programmes.

Figure 2 depicts the frequencies of the responses for activities that students would be interested in during AAI sessions. The top four responses were “petting and cuddles”, “playing with toys”, “dog-walking” and “quiet activities” such as studying or reading in the presence of the animal. Two “other” responses included animal yoga and working with rescue/shelter animals.

In terms of the potential benefits of AAIs (see Figure 3), students perceived AAIs as potentially enhancing their mood, providing comfort and well-being and reducing stress and loneliness. The reasons for attending an AAI session (see Figure 4) included experiencing stress (including during exam times), experiencing mental health difficulties, feeling lonely and experiencing personal crises.

#### 3.1.2. Student Support Services 

In collaboration with representatives of student support services, effective and sustainable implementation logistics were considered and decided upon. A referral route was designed through Student Well-being Advisors and mapped onto the referral process for student well-being support and access to student counselling. It was agreed that students could be referred to the programme if they met the criteria for requiring mental health support and were assessed as being at moderate risk of harm. Students who met the criteria of requiring well-being support should be referred rather than self-refer, to manage numbers and reach the students in most need. Engagement with Paws on Campus would not preclude students from participating in other support activities. The referral forms were aligned to Well-being Services risk assessment documentation for student well-being. Students from across the university could be referred (currently 49,000 students) and regular contact with Well-being Services leads was established alongside training sessions about the programme, referral forms and processes. These procedures were incorporated into the implementation materials to ensure adequate training of staff.

#### 3.1.3. Stakeholders for Animal Welfare

There was agreement that dogs would be most suited to a university AAI programme because of their behavioural characteristics and abilities, in line with previous research [10,11,12,13]. The programme is implemented by a team that includes an academic veterinarian, to ensure that canine health and welfare is central to the programme. The venue for the delivery of the programme is a veterinary outreach clinic run by the Royal (Dick) School of Veterinary Studies, University of Edinburgh, and was risk assessed as safe for dogs and people. It is within the central university campus, so highly accessible for students. The site was visited by an independent researcher and implementer of campus-based AAIs to evaluate and confirm the suitability of the space for its outlined purposes. Canine Concern Scotland agreed to partner to create the programme and supply registered Therapet dogs who were checked for temperament and health and fully insured for AAI work. A team of volunteers and their dogs was created in the local area. As a charity, they do not charge for this service, but require a donation. Canine welfare was agreed to be at the heart of the programme, both in terms of programme delivery and programme content. Sessions are timed to be one hour long for canine welfare, comfortable bedding for dogs is provided and there is a quiet space to which dogs can retire to ensure that the dogs are consenting to participate in activities. The Scottish SPCA agreed to contribute to the oversight of canine welfare in the programme.

#### 3.1.4. Creation of an Advisory Group

An advisory group was established to oversee the work of Paws on Campus, comprising university Well-being Advisor leads, veterinarians, a Scottish SPCA representative and student representatives.

### 3.2. Stage 2: Student Co-Production Workshops

#### 3.2.1. Workshop 1: Theory and Goals

This workshop allowed the researchers to understand students’ perceptions of the potential issues to be targeted by and the benefits of AAI. Table 3 summarizes the main themes from the group discussions.

The worksheets used in the workshop provided fine-grained feedback on how students thought an AAI could be implemented. Some questions had a range of responses; for example, the suggested ratio of students to dogs ranged from 2:1 to 4:1, and the number of sessions ranged between four and 10. Other questions reached consensus; for example, students agreed that session should last around one hour, and that a central goal for the programme should be reducing stress. They also agreed that a referral route through student support was important and highlighted the importance of having procedures in place to ensure dog welfare (e.g., allowing the dog to leave if they were uncomfortable). 

#### 3.2.2. Workshop 2: Activities and Themes

This workshop provided feedback on the proposed activities and generated ideas on how to improve them. Table 4 shows the feedback received for the activities trialled, including how students rated their own enjoyment of the activity and the dogs’ enjoyment of the activity. The students’ enjoyment of an activity was related to how much the dogs enjoyed it. For example, mindful petting and dog-training activities were very well received, as the dogs were relaxed and engaged. Students also brainstormed some new activities. A key theme that emerged was that unstructured activities were important. For example, students noticed the effect of the dog on the dynamics of the group, and that the dog was helping them to “put themselves out there”, encouraging feelings of acceptance and “going with the flow” socially. Students also enjoyed the chance to have free play with the dogs and a chance to just reflect on this, which highlights the importance of having an intervention which balances structured and didactic activities with more spontaneous unstructured activities. 

#### 3.2.3. Workshop 3: Session Content, Order, and Layout

The third workshop confirmed the key components of the programme: the number, order and content of sessions, and the materials and room setup. The group consensus was that four or five sessions was optimal, as this fits neatly into half a university semester. Discussion and feedback ensured coherent links between the session themes (psychoeducation), grounding exercises and dog activities. Students also discussed the feasibility of an outdoor session on learning through play and nature connectedness. In terms of the room setup, general feedback included the importance of adding window blinds to ensure the space felt safe and private, the option to sit on either the floor (on pillows) or on chairs and the importance of maximising floor space to allow for free interactions with the dogs. Figure 5 shows a diagram of the room setup that students and handlers agreed felt comfortable for both them and the dogs.

#### 3.2.4. Workshop 4: Trialling and Outdoor Session

The final workshop was an opportunity for the six post-graduates to trial a full session to get feedback on the flow, structure and session materials such as the booklet, as well as for trialling whether an outdoor session was feasible. Feedback on the booklets suggested that students appreciated lighter text and large diagrams, as well as a space within the booklet where they could write their reflections. Students did not want the booklets to feel too academic.

### 3.3. Stage 3: Paws on Campus Prototype and Feedback

The following Paws on Campus features were co-produced. Based on student survey results, Paws on Campus is a programme of four one-hour structured sessions with an optional fifth outdoor session. The intervention is not possible without the presence of dogs and psychoeducational content is informed by human mental health research and animal welfare science. The psychoeducational themes were based on the key issues faced by students, identified by students in the survey and in the co-production workshops. In line with the student survey findings, the programme is completed in small groups rather than individually to enhance social support and the sense of community. Student well-being staff consultations, student survey and co-production workshops had highlighted the need to target students experiencing mental health difficulties, so a referral process was established in collaboration with Well-being Advisors to ensure that students invited into the programme were experiencing mental health difficulties. This helped to target the programme to students most in need and to manage numbers. Students eligible for referral to Paws on Campus are assessed as being of moderate risk and experiencing mental health difficulties, including suicidal ideation with no plan of action. Following advice from animal welfare specialists, and in collaboration with the Royal (Dick) School of Veterinary Studies, sessions are held in a risk-assessed on-campus university community veterinarian clinic, which is safe for people and dogs and has a large room for group activities. The clinic has a small adjoining room that dogs can retreat to if they do not wish to participate in activities to ensure that we engage in canine consent, thereby promoting animal welfare. To ensure that the dogs engaged in the intervention were well suited to the role and would enjoy the activities, we collaborate with Canine Concern Scotland and a group of their registered volunteer Therapets dogs and handlers. The dogs have been health- and temperament-checked, have good basic training and are allowed to engage in natural dog behaviour off-lead within sessions to interact with students. They are also insured through Canine Concern Scotland for AAI work. Each session is led by a facilitator, trained in the programme, who is an academic with extensive experience of working with students. The programme is manualised and structured, although it is designed to be agile to be responsive to the individual needs of the students involved. Each session has a standard structure around four core elements:Grounding/mindfulness exercise (a different one each week to build a toolkit for students to practice and use later) [29].Psychoeducational content (an overview of psychological principles to enhance mental health and psychological literacy and aid understanding of canine welfare and care needs) [30,31].Purposeful activities with dogs (throughout sessions, students can interact with the dogs, but each week involves specific activities to link with the psychological theme of the week) [10,12,13].Group discussion and synthesis of the themes of the week to provide social support [32].

Weekly sessions build a spiral curriculum and integrate human well-being and canine welfare to form a One Health approach where the welfare of the dogs and the well-being of the students were equally important.

Thoughts and Feelings: This session focuses on the connections between human thoughts, feelings, bodily sensations and behaviours, and introduces some basic cognitive behavioural therapy principles [33]. It considers human reactions to different situations, and how our thoughts can link to emotions, sensations, behaviours, and the physiological impact of stress. The session also considers canine stress indicators [34], canine consent and individual differences among the dogs engaged in the session. The activities with the dogs involve reading the dogs’ body language, considering their mental states, engaging in canine consent activities and mindful interactions. This forms a foundation for interacting with the dogs in following sessions.Well-being and Welfare: This session introduces students to a range of classic and contemporary models of human needs and domains of human well-being [35]. It then considers theories of animal welfare freedoms and welfare domains [36]. The aim is to understand some of the factors that influence positive well-being/welfare. The activities with the dogs involve interacting with them and discussing with handlers the individual needs of specific dogs, how these needs are met and any challenges they face.Care and Compassion: This session invites students to consider how welfare and well-being needs might be met through compassion and care. The session introduces students to theories of compassion and self-compassion [37] and explores how interacting with animals can provide opportunities to give and receive care [38]. The activities with the dogs involve interacting with the dogs to provide care and receive non-judgmental support; we also invite students to engage in non-judgmental listening with their group and the dogs.Problem Solving and Help Seeking: This session explores the complexity of student life and the importance of breaking down problems into small steps and reaching out for help when needed [9]. It considers the basics of problem-focused therapy and tools to enable students to avoid feelings of overwhelm [39]. The session includes a discussion of learning in dogs [40] and problem solving as a positive form of enrichment for dogs, engages dogs in puzzle solving and learning novel behaviours [41], and includes clicker training with students.Positive Learning and Play (optional outdoor session): This optional session is designed to be run outside in a secure garden setting. The psychoeducational themes are positive learning and play, and nature-connectedness and well-being [42]. The activities include observing dogs in more active play and exploration and engaging with the natural environment. The logic model for Paws on Campus can be found in Figure 6.

#### 3.3.1. Step 1: Feedback from Student Volunteers

Initially, student volunteers from Veterinary Medicine and Psychology engaged in trial sessions. Four groups attended the Thoughts and Feelings session (N = 17), one group attended Well-being and Welfare (N = 5), one group attended Compassion and Care (N = 7), one group attended Communication and Body Language (N = 5) and one group attended Learning and Play (N = 5). After each session, students were asked feedback questions (see Table 5) and to anonymously rate the session on a five-point scale from strongly agree to strongly disagree as well as give text comments on the strengths and areas for improvement. All responses either agreed or strongly agreed with the statements, so Table 5 below reports the % that ‘strongly agreed’.

The ‘Best parts of the session’ open-ended comment box generated many comments from all participants in the following themes: (1) interacting with the dog, (2) calming down/de-stress, (3) grounding techniques/mindfulness, (4) positive mood/enjoyment, (5) positive environment/safe space, (6) listening to others/chatting to others/discussions and (7) session themes (e.g., well-being/welfare, compassion, learning). Example quotes include ‘I absolutely loved it!’ and ‘I enjoyed all of it’. The positive feedback suggested that the inclusion of grounding exercises, the mix of structured and unstructured activities with the dogs, the psychoeducational themes and the group context were all positive aspects of the programme, so these were retained.

The ‘Things that could be improved’ comment box generated few comments for changes but some additional positive feedback: Example quotes included ‘Longer sessions’, ‘nature sessions would be good’, ‘nil’, ‘literally nothing’ and ‘no improvements needed’. There were more suggested changes for the ‘Communication and Body Language’ session than for the other sessions; as a result, we revised this session to focus more on Problem Solving and Help Seeking, which had been highlighted as an area we could include in the programme.

#### 3.3.2. Step 2: Feedback from Students Experiencing Mental Health Difficulties

Based on the initial feedback and co-production workshops, we revised the ‘Communication and Body Language’ session to focus more on ‘Problem Solving and Help Seeking’ and developed the ‘Learning and Play’ session to be an optional outdoor session. We then tested the revised sessions with students referred from Well-being Services because they were struggling with mental health issues. Seven students engaged in the sessions, six gave consent to be interviewed, and one student attended less than three sessions so was not invited to participate in the feedback interview. Five students participated in feedback interviews about the programme.

Mood-check data involved participants completing a simple checklist at the start and the end of each session rating their momentary happiness, calmness, stress and depression (on a 10-point Likert scale). Paired-samples *t*-tests were employed to investigate whether pre-session to post-session changes in happiness, calmness, stress and depression were statistically significant across all the sessions taken by participants. The results indicated that happiness post-session (M = 6.35, SD = 1.66) was significantly higher than it was pre-session (M = 5.25, SD = 2.29) *t*(19) = 2.42, *p* = 0.026. Stress post-session (M = 5.05, SD = 1.67) was significantly lower than it was pre-session (M = 7.04, SD = 1.58) *t*(19) = −8.75, *p* < 0.001. Calmness post-session (M = 6.59, SD = 1.86) was significantly higher than it was pre-session (M = 4.05, SD = 1.35) *t*(19) = −4.93, *p* <.001. Depression post-session (M = 4.60, SD = 1.47) was significantly lower than it was pre-session (M = 5.50, SD = 2.40) *t*(19) = −2.20, *p* = 0.041. These results indicated that the sessions were having a positive impact on momentary mood, anxiety and stress levels, so the content and structure of the programme were having a positive impact on student well-being.

Five participants completed semi-structured interviews about their experiences of the programme, focusing on strengths, areas for improvements, outcomes they had experienced and their views on why the programme had influenced them. Thematic analysis was used to derive key themes, as illustrated in Figure 7. All participants perceived the programme as having a positive impact on their mental health. The key drivers for this change were perceived to be (1) practical mental well-being skills derived from grounding exercises and psychoeducation, (2) the therapeutic effects of interacting with dogs and (3) the connection and community fostered by working in small groups with university staff and Therapets volunteers. This feedback provided initial evidence of acceptability of the programme for the target group and further evidence that the key elements (grounding exercises, psychoeducational themes, interactions with the dogs and group setting), were all contributing to positive outcomes. Consequently, the key elements and themes were all retained in the programme.

The feedback and attendance records revealed that most students could only attend four sessions out of the five on offer, confirming the final programme structure of four sessions with an optional fifth by agreement with all involved and if weather permitted outdoor activities.

### 3.4. Academic Timetable and Delivery Implications

An important consideration regarding the structure and timing of Paws on Campus was the university academic calendar. It was decided to align Paws on Campus with semester times. The academic year is divided into two semesters, each comprising two 5-week teaching blocks followed by a revision period and assignment/exam weeks. Paws on Campus is timed to start in Week 2 of each teaching block in both semesters. This avoids revision and exam times and is easy for students to remember. As many students leave the university during the summer, Paws on Campus is not currently offered during the summer break. Sessions are held on Wednesday afternoons, which are university-wide teaching free times, so that Paws on Campus is accessible to students in need across the university.

The referral process from Well-being Services has generated referrals from all three colleges across the university (‘Medicine and Veterinary Medicine’, ‘Science and Engineering’ and ‘Arts, Humanities and Social Sciences’) and ongoing communications between the Paws on Campus team and Well-being Services leadership and staff. Referrals to Paws on Campus are continuing, enabling feasibility testing, prior to a full efficacy study. The aim is to embed Paws on Campus into student support practice in the university, while also contributing to international research on university AAIs.

### 3.5. Summary of Findings

A summary of key findings for each co-production stage is presented in Table 6.

## 4. Discussion

This study used a three-stage model of co-producing health interventions [21] applied to developing a university-based AAI for student mental health. To our knowledge, this is the first study to examine this process for AAIs conducted on college campuses. The iterative process took three years in total and involved engaging with university students, university student support staff and academic colleagues (e.g., ethical review), and non-academic animal welfare collaborators.

Engaging with a range of stakeholders from the beginning and involving them in programme co-production has led to a novel programme that meets the needs of vulnerable students, is embedded within the university’s student support services and has established ongoing collaborations with Canine Concern Scotland’s Therapets Service and the Scottish SPCA.

The process was iterative and time-consuming but worked well to create the relationships required for programme delivery and was fruitful in leading to the development of a new canine-facilitated programme for student mental health.

### 4.1. Paws on Campus Programme

The resulting One Health programme is manualised and replicable, focusing on student mental health and canine welfare. Rather than being conceptualised as a time-limited research programme, as many campus-based AAIs are [12], it has been developed as a student support programme with an aligned research programme. This will ensure it can be run in the long-term because it is embedded within the university student support structures, while also contributing to international research on university AAIs.

Paws on Campus builds on the evidence base of existing campus-based AAIs, focusing on student well-being [14,18], stress reduction [13] and employing group contexts [43]; however, the co-production process has added novel elements that make the programme unique.

Firstly, Paws on Campus focuses on supporting vulnerable students who have been assessed as in need of mental health support. This was facilitated by the team including psychologists whose research interests include developmental processes in mental health and human–animal interactions. This expertise enabled the team to draw from research on psychological therapies to inform the psychoeducational aspects of the programme. It was also essential to engage the expertise of student support services in co-producing the programme because they have expertise in and practical experience of supporting students with mental health difficulties. While many campus dog programmes engage students at times of heightened stress, such as exam time [13], most are not targeted specifically at students who are experiencing mental health issues.

Secondly, Paws on Campus is a programme of four linked weekly sessions, rather than one-off drop-in sessions [44], with a structured curriculum. While other AAI programmes have involved a series of sessions, these have often been mental health interventions with dogs added to test the impact of AAIs [45,46], rather than specifically designed One Health AAIs. Where possible, students meet in the same small groups, with the same facilitator and dogs with their handlers, which enables not only a spiral curriculum to be employed [47] but for relationships to build between students, staff and the dogs they meet on a weekly basis. Throughout all sessions, there are opportunities for off-leash unstructured interactions with the dogs, both as an ice breaker at the beginning of sessions and throughout the activities. Spontaneous ‘interruptions’ from the dogs engaging with the students aids the flow of the sessions by injecting surprise and humour and facilitating laughter and relaxation. Spontaneous actions by the dogs often lead into interesting discussions and the sharing of information by the participants. Sometimes, when students were interacting with the dogs and focusing on the dog, they share information about challenging situations or how they were feeling.

Thirdly, Paws on Campus is embedded within the university context ensuring that it complements both student well-being services and academic activities within the university. It is available to all students across the university who meet the inclusion criteria of requiring well-being support services for their mental health difficulties. The programme is aligned to the academic timetable in terms of semester times and avoiding exams times, but also in being held during weekly non-teaching times to enable student access. There is also a fit with student support services, so that students who are referred to Paws on Campus maintain contact with their Well-being Advisor during and after participating in the programme. This ensures vulnerable students are fully supported in the university.

Fourthly, as a One Health intervention, canine welfare is as important as human well-being and central to both the content of the intervention sessions and the delivery of the programme. Sessions are timed for one hour and the venue is safe for dogs and people, canine consent is taught and employed in sessions and there is a breakout space for dogs to ensure positive welfare [48]. The human–animal interaction activities are designed to be enjoyable for dogs and students, often involving canine enrichment activities (e.g., dog puzzles) [49], and dogs are monitored for signs of stress by all involved. Paws on Campus explicitly avoids instrumentalising the dogs and treating them as ‘a resource’. As well as focusing on human well-being, students are taught about animal welfare needs and to respect the needs and wishes of the dogs in the sessions. Students learn about the dogs’ individual characteristics and histories (e.g., rehoming history) and often draw parallels between their own mental health and the well-being of the dogs. Taking a One Health approach requires that the animals who participate in AAIs are not viewed as a resource that can be used without consideration of their welfare. As the human–animal bond is reciprocal, AAIs are likely to be more effective in supporting human health and well-being if the animals involved are respected, consent to the activities and enjoy positive welfare.

### 4.2. Benefits of Co-Producing an AAI for Mental Health

There have been a range of benefits of taking a co-production approach to Paws on Campus. By engaging students in a variety of activities in the three stages of co-production, we have developed an intervention that meets their needs, is attractive to them and enjoys high demand and good attendance. Importantly, including Well-being Services leads and advisors in co-production and having academic staff as leaders has ensured that the programme is embedded within the university’s academic timetable and support services. This means that vulnerable students can be referred to the programme and maintain well-being support after programme completion, and that the programme and sessions fit with the teaching timetable and avoid exams and holidays. This has also enabled the programme to be developed as a student support service, but with planned continuous evaluation and feedback to contribute to academic research on AAIs. Taking a One Health approach and involving veterinarians and animal welfare specialists has been vital to ensure that we have a non-academic partner who can provide access to temperament- and health-checked dogs and their handlers, and that the activities have been developed with positive dog welfare central to the programme content and delivery.

### 4.3. Limitations and Future Directions

Developing a complex One Health intervention for vulnerable students is not without challenges. The staff and students, and the non-academic collaborators who participated, were self-selected and interested in dogs and AAIs, so are not necessarily representative of their peers. Developing a programme to fit within specific university contexts may impact the generalisability of the programme for use in other institutions. It should also be noted that this co-production work has not tested the efficacy of the intervention, that no control group was included and that the sample size was small. Currently, feasibility assessments are being carried out to further examine effectiveness and implementation issues. Future research should include a randomised controlled trial to test efficacy and implementation work to adapt Paws on Campus for delivery in other university contexts. The programme has been created in the UK, with UK higher education policies and practices and UK animal welfare legislation. It is important that high animal welfare standards are adhered to if the programme is implemented in countries with fewer welfare protections.

## 5. Conclusions

Through a three-stage co-production process, we have created a novel One Health University AAI, Paws on Campus. The co-production process led to a unique programme that builds upon existing evidence, is suited to the needs of vulnerable students, and prioritizes canine welfare. Paws on Campus is a targeted AAI involving a series of four weekly sessions with a spiral curriculum. The next steps of the work are a feasibility study and then a randomised controlled trial to test efficacy.

## Figures and Tables

**Figure 1 ijerph-21-01066-f001:**
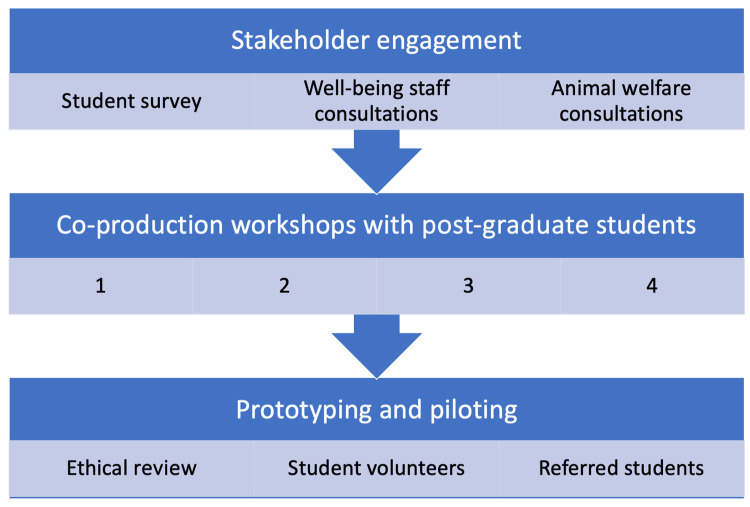
Structure of co-production processes.

**Figure 2 ijerph-21-01066-f002:**
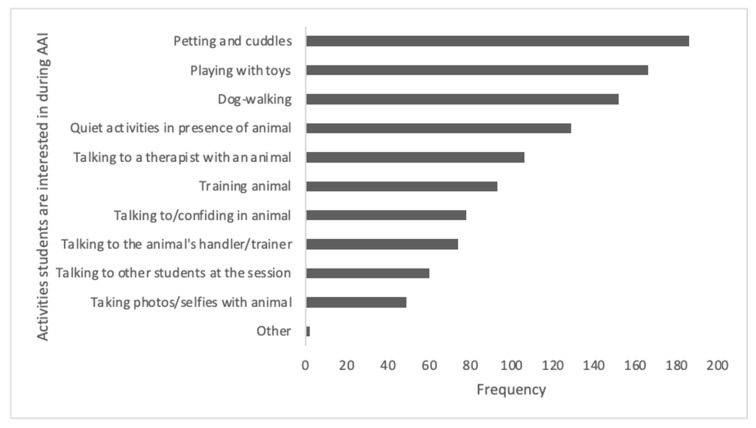
Frequencies of activities students would be interested in during an AAI session.

**Figure 3 ijerph-21-01066-f003:**
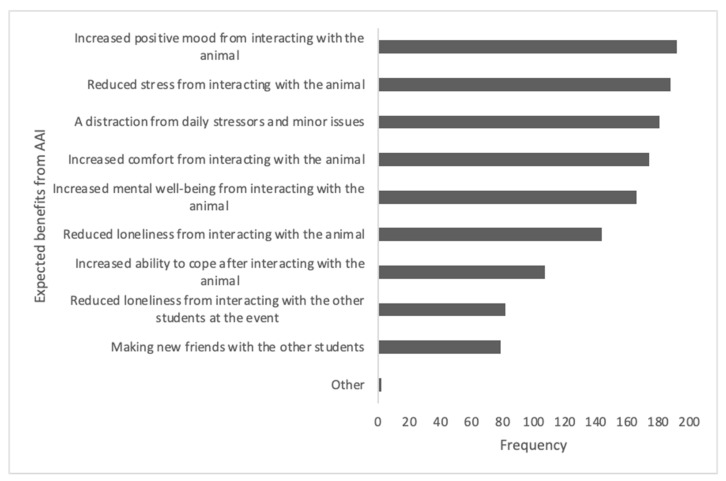
Frequencies of expected benefits from attending an AAI session.

**Figure 4 ijerph-21-01066-f004:**
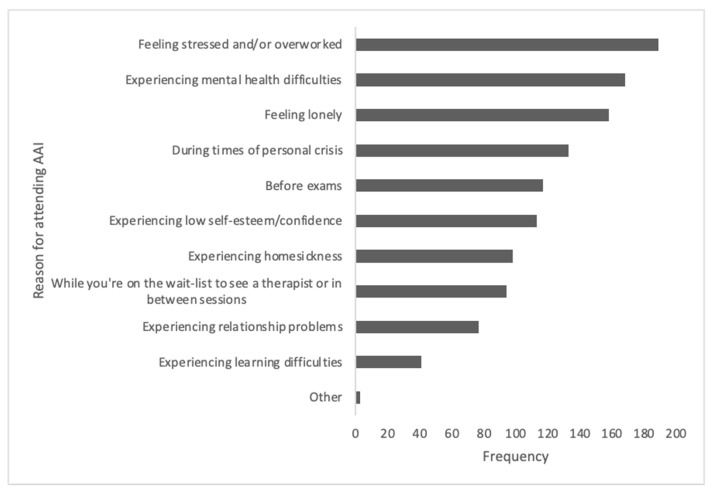
Frequencies of reasons for attending an AAI session.

**Figure 5 ijerph-21-01066-f005:**
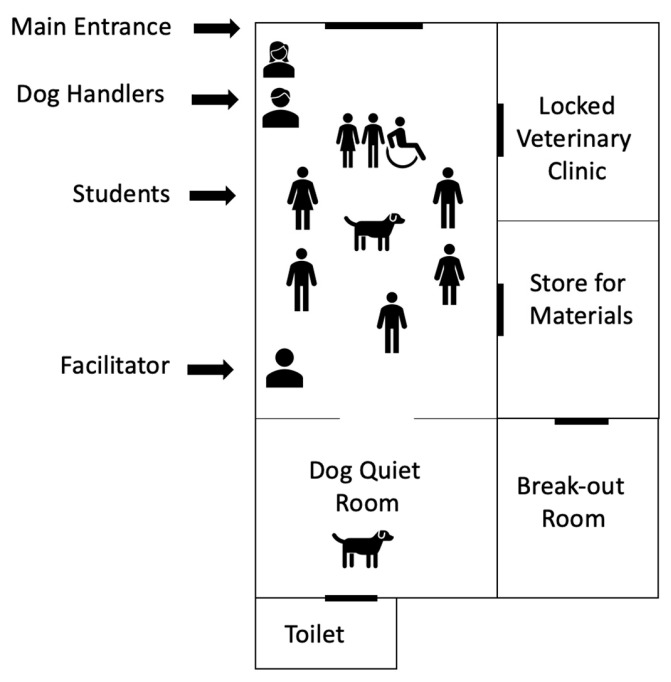
Diagram of setup of Paws on Campus venue.

**Figure 6 ijerph-21-01066-f006:**
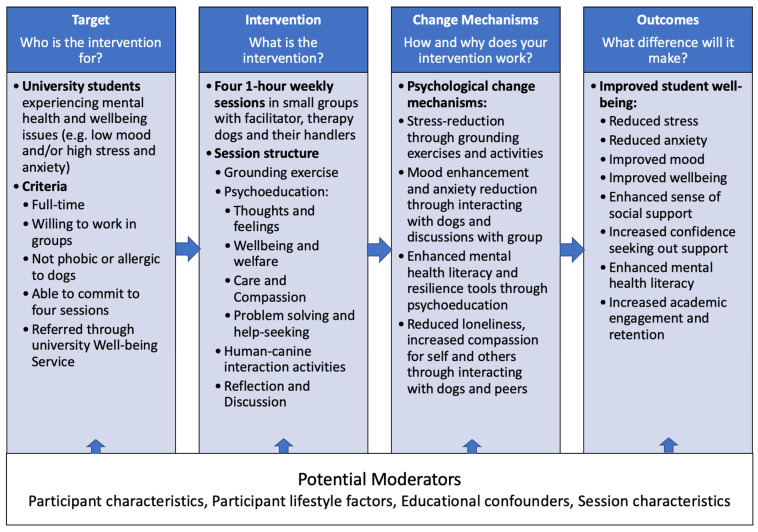
Logic model of Paws on Campus.

**Figure 7 ijerph-21-01066-f007:**
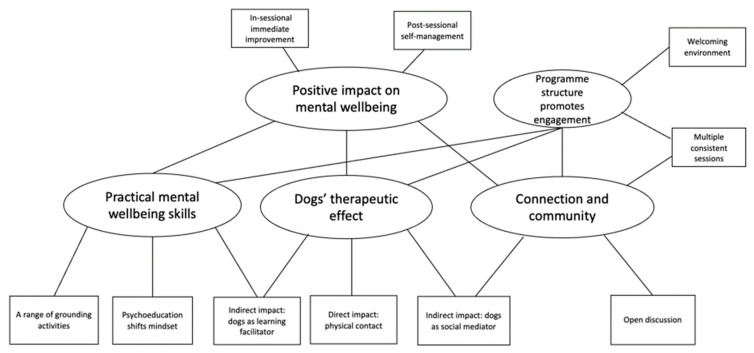
Thematic map of Paws on Campus’ impact on participants.

**Table 1 ijerph-21-01066-t001:** Demographics of the participants who completed the student survey, N = 204.

Variable	Frequency	%
Age		
18–21	46	22.5
22–25	87	42.6
26–29	36	17.6
30+	35	17.2
Gender		
Female	179	87.7
Male	22	10.8
Non-binary	2	1.0
Prefer not to say	1	0.5
University Degree		
Undergraduate	83	40.7
Post-graduate Taught	76	37.3
Post-graduate Research	45	22.1
Degree Programme		
Arts	17	8.3
Humanities	20	9.8
Medicine	5	2.5
Science	29	14.2
Social Sciences	85	41.7
Engineering or Mathematics	10	4.9
Veterinary Medicine	17	8.3
Business	9	4.4
Other Healthcare	6	2.9
Other	6	2.9
Location of University		
United Kingdom	135	66.2
Canada	21	10.3
United States of America	29	14.2
Australia	5	2.5
Europe	11	5.3
Asia	2	1.0
Uruguay	1	0.5
International Student		
No	142	70
Yes	61	30

**Table 2 ijerph-21-01066-t002:** Descriptive statistics regarding student AAI preferences, N = 204.

Variable	Frequency	%
Previously attended an AAI session at university		
Yes	73	36.0
No	130	64.0
Interested in attending an AAI session if offered at university		
Yes	153	75.0
No	20	9.8
Maybe	31	15.2
What animals would you want at an AAI session? (multiple answers allowed)		
Dogs	185	90.7
Cats	128	62.7
Rabbits	97	47.5
Guinea pigs	77	37.7
Miniature horses/Shetland ponies	103	50.5
Other	33	16.2
How would you prefer the AAI session to be run? (multiple answers allowed)		
One-on-one with animal and therapist	114	55.9
One-on-one with animal only	120	58.8
In small groups with animal and other students	130	63.7
In large groups with animal and other students	10	4.9
How often would you attend an AAI at your university?		
Never/unlikely to attend	11	5.4
One time only	28	13.7
Monthly	88	43.1
Weekly, if offered	77	37.7
What time of day would you be most like to attend an AAI? (multiple answers allowed)		
Mornings	70	34.3
Afternoons	157	77.0
Evenings	95	46.6
Weekends only	37	18.1

**Table 3 ijerph-21-01066-t003:** Students’ perceptions of issues addressed by and benefits of campus dog programmes.

What Are Student-Focused Problems?	How Can Campus Dog Programmes Help?
**Mental Health** Stress Depression Anxiety Coping mechanisms **Social Issues** Loneliness Communication Balancing academic and social life **Self-care** Navigating adulthood Work–life balance Taking time for mindfulness	**Mental Health** Stress reduction Mood improvement Short-term anxiety reduction They provide a safe space **Social Issues** Promote connections (human–human and human–dog) Promote social engagement **Self-care** Grounding techniques Knowledge and skills Chance to practice mindfulness **Animal Welfare** Chance to learn about dog welfare Learning about dog communication

**Table 4 ijerph-21-01066-t004:** Feedback on activities trialled in Co-production Workshop 2.

Activity	Student Enjoyment (Mean)	Dog Enjoyment (Mean)	Additional Notes
Dog tricks	9.5	9	Have suggested tricks to do with dogs, especially as a starting point
Mindful petting	10	10	Letting the dog guide the interaction was important and helpful
Problem solving	9.5	9	Fun, but can be tricky to get the difficulty of the problem just right (not too hard, not too easy!)
Dog eye gaze	8	7	Eye gaze felt a bit stressful for the dogs if prolonged—short gazes worked well

**Table 5 ijerph-21-01066-t005:** Percentage of students who strongly agreed with feedback statements.

	Thoughts and Feelings	Well-Being and Welfare	Compassion and Care	Communication and Body Language	Learning and Play
I enjoyed the session	88.2%	100%	100%	100%	100%
I enjoyed interacting with the dog	100%	100%	100%	100%	100%
I found the session helpful	64%	100%	100%	80%	60%
I found interacting with the dog helpful	88.2%	100%	100%	100%	80%
Meeting in a group was a positive experience	47%	100%	85.7%	80%	60%
I would recommend POC to others	94.1%	100%	100%	100%	100%
I would like to do more POC sessions	100%	100%	100%	100%	100%
The POC venue was good	64%	N/A	N/A	N/A	N/A
POC is good for students	88.2%	N/A	N/A	N/A	N/A

**Table 6 ijerph-21-01066-t006:** Summary of findings of the 3-stage co-production of Paws on Campus.

Activity	Objectives	Summary Results
Stage 1: Stakeholder Consultation		
Student survey	Identify student AAI preferences and expectations	There was high interest in AAI Preference for interactions with dogs Small group sessions Monthly or weekly sessions Afternoons Include a range of activities and interaction opportunities Targeting students experiencing stress, mental health issues and loneliness
Well-being Advisor consultations	Identify student referral route and embed within university student support systems	The programme would be embedded within university student support systems and available to vulnerable students across the university Students who meet the criteria of requiring well-being support should be referred rather than self-refer, to manage numbers and reach students in most need Students should maintain contact with their named Well-being Advisor throughout Paws on Campus to ensure continuous support Engagement with Paws on Campus would not preclude students from other support activities
Consultations with animal welfare professionals	Establish best animal welfare practice for university AAI	Enthusiasm and commitment for the programme Non-academic partner Canine Concern Scotland’s Therapets Service and team of volunteer handlers and their dogs Access to the ‘Dick Vet in the Community Clinic’ was granted for programme delivery
Stage 2: Co-production Workshops		
Four workshops with students	Agree on the core components of the intervention to enhance student well-being through AAI	Focus on alleviating student anxiety, depression, stress and loneliness among vulnerable students Programme of four to five sessions Session structure including grounding exercise, psychoeducation theme and canine–human interaction activities Session themes and activities
Stage 3: Prototyping		
Prototype	Create manualised intervention with all implementation materials including referral process and evaluation tools	Set of four sessions, following spiral curriculum focusing on canine welfare and human well-being Standardised structure for sessions including grounding exercises, psychoeducation and human–canine interaction activities
Ethical review of all materials by human and animal health experts	Identify ethical issues with intervention and feedback materials	Ethical approval was granted for intervention materials and feasibility piloting
Student volunteer sessions	Initial feedback on sessions	High positivity about session focus and structure Narrowing down of some sessions to focus on students concerns
Referred vulnerable student feedback	Test implementation processes and gain feedback from target group	Attendance was good and engagement high A four-session programme is optimal Evidence of mood increases and stress and anxiety reduction following sessions Interviews highlighted value of interactions with dogs, grounding exercises and psychoeducation and sense of community in groups

## Data Availability

Data may be available upon request to authors.

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
