# Peer review of "Co-Producing Paws on Campus: A Psychoeducational Dog-Facilitated Programme for University Students Experiencing Mental Health Difficulties"

_ijerph, 2024, doi:10.3390/ijerph21081066_

Round 1

Reviewer 1 Report

Comments and Suggestions for Authors

The paper is of interest and value to a wide audience including animal welfare professionals, and higher education providers. There are some minor grammatical errors throughout. A thorough proof-read would be advisable. 

I have included some specific feedback below (with reference to article line number where appropriate). 

Line 163 - "The university already runs..." - up until this point, there narrative hasn't been around a specific facility. Some clarification around this would help (e.g. The university the study was undertaken at...)

Line 176 - change 'vets' to 'veterinarians' - note, this change needs to occur throughout the paper. 

Line 191/200 - reference to student but lack of clarity on which students - is this the six PG students, or the student volunteers. Specifiying each time would help with clarity. 

Line 214 - reference to 'we' - change to third person to be consistent with remainder of paper

Line 222/308/ - when numbers are 0-9, write in full. 

Figure 1 - need clarity around which students the workshops were with - PG students/volunteers?

Line 248 - remove 'rather'

Line 308 - change 'really useful' as this is too informal in comparison to the rest of the paper. Consider 'This sessions allowed the researchers to...'

Line 318 - Change 'importance having' to 'importance of having'

Table 4 - change 'possible' to 'possibly' ,or remove word all together

Line 356 - 'whether an outdoor session' - sentence needs finishing

Line 358 - 'diagram' becomes 'diagrams'

Line 358 - 'whether' becomes 'where'

Line 369 - 'help' becomes 'held'

Line 369 - Is the community vet clinic on-campus? Early reference to on-campus clinic, so if so, please clarify here

Line 401 - change 'we' to 'Researchers' (see line 409/413/420 also)

Figure 6 - still contains editing elements.

Line 459/460 - sentence makes no sense

Line 489 - diets? wrong word

Line 510 - change 'used three' to 'used a three'

Line 533 - change 'but' to 'however'

Line 563 - change 'we'

Line 564 - 'terms semester times' - makes no sense

Line 574 - change 'to enjoyable' to 'to be enjoyable'

General: Readability of the results section is difficult, with it hard to know what was happening at each stage, who was involved and in what order. Maybe consider a additional diagram/flow chart to explain this. 

Comments on the Quality of English Language

English language quality is exceptions, barring some minor grammatical errors. 

Author Response

Dear Reviewer 1, Thank you for your thoughtful comments on our paper. We appreciate your time and thorough review you have provided. We have made all the changes you have suggested, including changes to figures to enhance readability. We have also proof-read the paper to correct grammatical errors.

General Comment

The paper is of interest and value to a wide audience including animal welfare professionals, and higher education providers. There are some minor grammatical errors throughout. A thorough proof-read would be advisable. I have included some specific feedback below (with reference to article line number where appropriate).

Comment 1: Line 163 - "The university already runs..." - up until this point, there narrative hasn't been around a specific facility. Some clarification around this would help (e.g. The university the study was undertaken at...)

Response 1: We have adopted the suggested wording.

Comment 2: Line 176 - change 'vets' to 'veterinarians' - note, this change needs to occur throughout the paper.

Response 2: We have changed vets to veterinarians throughout the paper.

Comment 3: Line 191/200 - reference to student but lack of clarity on which students - is this the six PG students, or the student volunteers. Specifiying each time would help with clarity.

Response 3: We have edited Figure 1 to make it clear that post-graduate students were involved in the co-production workshops.

Comment 4: Line 214 - reference to 'we' - change to third person to be consistent with remainder of paper

Response 4: We have changed we to ‘it was’.

Comment 5: Line 222/308/ - when numbers are 0-9, write in full.

Response 5: We have changed numerals 0-9 to words where they are not numerals in titles (e.g., titles of figures, tables, or activities).

Comment 6: Figure 1 - need clarity around which students the workshops were with - PG students/volunteers?

Response 6: We have clarified in the text that the co-production workshops involved six post-graduate students.

Comment 7: Line 248 - remove 'rather'

Response 7: We have removed ‘rather’

Comment 8: Line 308 - change 'really useful' as this is too informal in comparison to the rest of the paper. Consider 'This sessions allowed the researchers to...'

Response 8: We have changed the text to: ‘This session allowed the researchers to understand students’ perceptions of the potential issues targeted by and benefits of campus dog programmes.’

Comment 9: Line 318 - Change 'importance having' to 'importance of having'

Response 9: We have added ‘of’

Comment 10: Table 4 - change 'possible' to 'possibly' ,or remove word all together

Response 10: We have removed the word ‘possible’

Comment 11: Line 356 - 'whether an outdoor session' - sentence needs finishing

Response 11: Added text ‘was feasible’.

Comment 12: Line 358 - 'diagram' becomes 'diagrams'

Response 12: Changed ‘diagram’ to ‘diagrams’.

Comment 13: Line 358 - 'whether' becomes 'where'

Response 13: Changed 'whether' to 'where'.

Comment 14: Line 369 - 'help' becomes 'held'

Response 14: Changed ‘help’ to held’.

Comment 15: Line 369 - Is the community vet clinic on-campus? Early reference to on-campus clinic, so if so, please clarify here

Response 15: Added in ‘on-campus university’ to clarify the clinic is within the university

Comment 16: Line 401 - change 'we' to 'Researchers' (see line 409/413/420 also)

Response 16: Changes text to ‘The session’ because this refers to the content of the session. We have made minor edits to other parts of this text for consistency wit this change.

Comment 17: Figure 6 - still contains editing elements.

Response 17: Figure 6 has been revised and simplified to enhance readability.

Comment 18: Line 459/460 - sentence makes no sense

Response 18: We have changed to: ‘Five students participated in feedback interviews on the programme’.

Comment 19: Line 489 - diets? wrong word

Response 19: We have changed ‘diets’ to ‘weeks’. The university in which the research is based uses the term ‘diet’ to mean a period of time (e.g. exam period), but we accept this is not a common usage of the word.

Comment 20: Line 510 - change 'used three' to 'used a three'

Response 20: Changes text to 'used a three'

Comment 21: Line 533 - change 'but' to 'however'

Response 21: Changed 'but' to 'however'

Comment 22: Line 563 - change 'we'

Response 22: We have removed ‘We ensured that’

Comment 23: Line 564 - 'terms semester times' - makes no sense

Response 23: We have revised the sentence to read: ‘The programme is aligned to the academic timetable in terms of semester times and avoiding exams times, but also by being held during weekly non-teaching times to enable student access.’

Comment 24: Line 574 - change 'to enjoyable' to 'to be enjoyable'

Response 24: Changed 'to enjoyable' to 'to be enjoyable'

Comment 25: General: Readability of the results section is difficult, with it hard to know what was happening at each stage, who was involved and in what order. Maybe consider a additional diagram/flow chart to explain this.

Response 25: We appreciate the multi-method approach to this co-production work is complex and the results are difficult to follow so we have added text in lines 246-248 to explain the organisation of the Result section. Figure 1 provides a diagram of the structure of the results section.

Comment 26: English language quality is exceptions, barring some minor grammatical errors.

Response 26: Thank you

Reviewer 2 Report

Comments and Suggestions for Authors

This report was interesting and extremely well written.  It clearly outlined the steps taken to design an AAI for university students with mental health issues.  I have only a few minor suggestions!

Line 24: Move AAIs after sustainable to help make this sentence clearer

Line 248: Avoid starting a sentence with a number

Figure 3-4: the text is just a little bit hard to read in places.  Consider changing font type or size to make it a little easier to read

Figure 5: Handwriting here is also hard to read with the size.  Could this be digitized to a typed/drawn diagram to make it easier to read?

Line 356-357: "as well as trialing whether an outdoor session"...  appears to be an incomplete sentence

Line 358: whether-->where

Line 370:help--> held

Figure 6: The light blue and green text text is a little hard to read (especially when printed in gray scale!!)

Line 489: exam diets???  What does this mean?

Figure 7: Can't read the small font

Line 564: terms [of] semestser

Line 574: designed to [be] enjoyable

Author Response

Dear Reviewer 2, Thank you for such a positive review of our work. We appreciate the time you have taken to go through our manuscript and have made all the changes you suggest, including editing the figures to enhance readability. 

Comments and Suggestions for Authors

This report was interesting and extremely well written. It clearly outlined the steps taken to design an AAI for university students with mental health issues. I have only a few minor suggestions!

Comment 1: Line 24: Move AAIs after sustainable to help make this sentence clearer

Response 1: We have moved AAIs to after sustainable

Comment 2: Line 248: Avoid starting a sentence with a number

Response 2: We have changed text to ‘Seventy seven percent’.

Comment 3: Figure 3-4: the text is just a little bit hard to read in places. Consider changing font type or size to make it a little easier to read

Response 3: We have increased to size of the figures to make them easier to read.

Comment 4: Figure 5: Handwriting here is also hard to read with the size. Could this be digitized to a typed/drawn diagram to make it easier to read?

Response 4: We have removed the drawing and inserted a diagram with typed labels to illustrate to Paws on Campus venue set up.

Comment 5: Line 356-357: "as well as trialing whether an outdoor session"... appears to be an incomplete sentence

Response 5: We have added ‘was feasible’ to complete the sentence.

Comment 6: Line 358: whether-->where

Response 6: We have changed ‘whether’ to ‘where’

Comment 7: Line 370:help--> held

Response 7: We have changed ‘help’ to ‘held’

Comment 8: Figure 6: The light blue and green text text is a little hard to read (especially when printed in gray scale!!)

Response 8: We have re-created the logic model using a simpler template to enhance readability.

Comment 9: Line 489: exam diets??? What does this mean?

Response 9: The university in which this research was carried out used the term ‘diet’ to mean time period (e.g. exam time). We have removed ‘diets’ and replaced with ‘weeks’.

Comment 10: Figure 7: Can't read the small font

Response 10: We have increased the size of this image to enhance readability.

Comment 11: Line 564: terms [of] semestser

Response 11: We have revised this sentence for clarity to: ‘The programme is aligned to the academic timetable in terms of semester times and avoiding exams times, but also by being held during weekly non-teaching times to enable student access.’

Comment 12: Line 574: designed to [be] enjoyable

Response 12: We had added ‘be’ in the text.

Reviewer 3 Report

Comments and Suggestions for Authors

Hello dears;

Thanks for this research.

Strengths:

The novelty of research, especially the help of animals in improving human mental health.

Comments:

1-Introduction : In expressing the importance of the topic, more attention has been paid to mental health problems and the increase in mental disorders in the target society, and the main issue, which issue the use of animals to help humans, has been less mentioned. 

2-In using pet therapy and AAIs, attention should be paid to choosing the type of pet( behavioral characteristics and abilities), but it is not mentioned in this research.

3- The sampling method  as well as the include and exclude criteria are not well defined.

4- Explain more about the use of mental health questionnaire and its validity and reliability.

5-The absence of a group control reduces the validity of the research.  

Author Response

Dear Reviewer 3, Thank you for your careful consideration of our manuscript.  We have made the changes you have suggested and clarified text where it was not salient enough.   We greatly appreciate the time you have taken to consider our work.

Hello dears; Thanks for this research.

Strengths: The novelty of research, especially the help of animals in improving human mental health.

Comments:

Comment 1: Introduction: In expressing the importance of the topic, more attention has been paid to mental health problems and the increase in mental disorders in the target society, and the main issue, which issue the use of animals to help humans, has been less mentioned.

Response 1: Thank you for raising this point.  We have included  reviews of the AAI evidence in the IntroductionSections 1.2, 1.3 and 1.4. We have provided more in-depth coverage of AAI research in Results Section 3.3. where we outline the Paws on Campus programme with relevant citations to previous research on AAI. The Introduction focused on the novel contributions of the work, which is the adoption of co-production methods to develop AAI programmes. As this is a long paper with limited space for expanding the text, we felt it was most important to focus on the original contributions of the work in the Introduction.

Comment 2: In using pet therapy and AAIs, attention should be paid to choosing the type of pet (behavioral characteristics and abilities), but it is not mentioned in this research.

Response 2: The choice of dogs was in part the result of the survey findings, where 91% of students stated their preference was for interacting with dogs (see Section 3.1.1). We have also expanded on the appropriateness of working with dogs in section 3.1.3 lines 427-429.

Comment 3: The sampling method as well as the include and exclude criteria are not well defined.

Response 3: We have provided sampling and inclusion and exclusion criteria in relation to each of the stages of co-production. The student survey sampling method is explained in Section 2.3.1. The student wellbeing staff sample is explained in 2.3.2. Section 2.3.3. explains the sampling method of the animal welfare professionals. Section 2.4 outlines the post-graduate student sample for the co-production workshops. Section 2.5 describes the sampling method for the prototyping and feedback stage including student volunteers and referred students. We have edited the text to clarify sampling methods and make them more salient.

Comment 4: Explain more about the use of mental health questionnaire and its validity and reliability.

Response 4: We have added information about the measures used in Methods Section 2.5 lines 231-333 and in Results Section 3.3.2.

Comment 5: The absence of a group control reduces the validity of the research.

Response 5: This research was not an evaluation study to test efficacy, so there is no control group. It is a co-production study, where we used a three-stage model to co-produce the intervention with a range of university students and professionals. The aim of this paper is to build upon public health co-design research approaches by applying them to developing an AAI for the first time. The paper shares with the international community the co-production process and design of the resulting Paws on Campus programme. Following UK Medical Research Council guidelines for developing and evaluating complex health interventions, this co-production work and prototype will be followed with a small-scale feasibility study (currently underway), and then a large-scale RCT efficacy trial which will involve a control group. Thank you for reminding us of the importance of control groups for testing efficacy.

Round 2

Reviewer 3 Report

Comments and Suggestions for Authors

Thanks for correcting the article.

Author Response

Comment 1: Thanks for correcting the article.

Response: Thank you for giving your time to review our revised manuscript and for providing such valuable comments on our work.